biomathematics/biomedical engineering/systems biology

measles, sliding mode control, biomedical applications of control engineering, morbillivirus infection, viral clearance, T-cell immunity

**Author for correspondence:**
Anet J. N. Anelone
e-mail: anet.anelone@sydney.edu.au

# Control theory helps to resolve the measles paradox

Anet J. N. Anelone[1], Edward J. Hancock[1,2], Nigel Klein[3], Peter Kim[1] and Sarah K. Spurgeon[4]

[1]School of Mathematics and Statistics, and [2]The Charles Perkins Centre, The University of Sydney, Camperdown, New South Wales 2006, Australia
[3]Department of Infection, Immunity and Inflammation, UCL Great Ormond Street Institute of Child Health, 30 Guilford Street, London WC1N 1EH, UK
[4]Department of Electronic and Electrical Engineering, University College London, Torrington Place, London WC1E 7JE, UK

 AJNA, 0000-0002-7481-8134; EJH, 0000-0003-1606-9611;
NK, 0000-0003-3925-9258; PK, 0000-0002-1492-4744;
SKS, 0000-0003-3451-0650

Measles virus (MV) is a highly contagious respiratory morbillivirus that results in many disabilities and deaths. A crucial challenge in studying MV infection is to understand the so-called 'measles paradox'—the progression of the infection to severe immunosuppression before clearance of acute viremia, which is also observed in canine distemper virus (CDV) infection. However, a lack of models that match *in vivo* data has restricted our understanding of this complex and counter-intuitive phenomenon. Recently, progress was made in the development of a model that fits data from acute measles infection in rhesus macaques. This progress motivates our investigations to gain additional insights from this model into the control mechanisms underlying the paradox. In this paper, we investigated analytical conditions determining the control and robustness of viral clearance for MV and CDV, to untangle complex feedback mechanisms underlying the dynamics of acute infections in their natural hosts. We applied control theory to this model to help resolve the measles paradox. We showed that immunosuppression is important to control and clear the virus. We also showed under which conditions T-cell killing becomes the primary mechanism for immunosuppression and viral clearance. Furthermore, we characterized robustness properties of T-cell immunity to explain similarities and differences in the control of MV and CDV. Together, our results are consistent with experimental data, advance understanding of control mechanisms of viral clearance across morbilliviruses, and will help inform the development of effective treatments. Further the analysis methods and results have the potential to advance understanding of immune system responses to a range of viral infections such as COVID-19.

# 1. Introduction

Measles virus (MV) is one of the most transmissible viruses affecting humans and results in many disabilities and deaths [1–3]. Measles infections caused more than 140 000 deaths in 2018 worldwide, mostly among children under five years old [2]. In 2019, there were over 400 000 confirmed cases reported to the World Health Organization; these cases strained healthcare systems, led to serious illness, disability and deaths in many parts of the world [1,4]. Vaccine coverage is still below the 95% needed to prevent outbreaks and there is no specific antiviral treatment [1–3]. Consequently, it is important to understand how measles is controlled *in vivo* to prevent, treat and manage infections.

A critical challenge in studying measles infection is to understand the so-called 'measles paradox'—the progression of the disease to severe immunosuppression before clearance of acute viremia. Typically, when viruses induce immunosuppression, this leads to high and persistent viral loads, immune dysfunctions and even death, as in HIV/AIDs or chronic lymphocytic choriomeningitis virus (LCMV) infection [5,6]. MV is a member of the morbillivirus genus which causes respiratory infections, and the measles paradox is also exhibited by other morbilliviruses such as canine distemper virus (CDV). Morbillivirus infections are atypical respiratory infections, where immune cells rather than respiratory epithelial cells are the main targets [7–9]. MV, CDV and other morbilliviruses preferentially infect dendritic cells and white blood cells expressing the receptor CD150 or signalling lymphocyte activation molecule F1 (SLAM/F1), such as B cells and T cells [7–9]. When a large number of immune cells are lost, the host becomes vulnerable to opportunistic infections which may lead to disabilities or deaths [2,3,10,11]. The period of immunosuppression is followed by the paradoxical clearance of acute viremia [10,12–15]. The magnitude and timing of immunosuppression and viral clearance vary for different morbilliviruses and between individuals infected by a given morbillivirus [9,10,14,15]. Since the virus mainly replicates in lymphoid tissues, virus-induced lymphocyte death seems to be the major cause of immunosuppression and extensive target-cell depletion could lead to viral clearance [8,9,14,16]. Fatal immunosuppression has been observed in dogs and ferrets infected by CDV despite a partial protection from CD8+ T cells [14,15,17]. Some CDV-infected dogs experienced a peak in immune lymphocyte-mediated cytotoxicity (ILMC) between 21 and 28 days post infection [14]. Measles infection leads to a large increase in MV-specific T cells in animal models of measles infection in humans [10]. This result and others suggest that virus-specific T-cell immunity plays a major role in typical cases of measles infection [10,12,16,18]. An important part of this puzzle is to determine mechanisms underlying differences in immunosuppression and viral clearance.

However, a lack of progress in mathematical modelling has restricted our understanding of the within-host dynamics and measles paradox. Mathematical modelling has delivered significant qualitative and quantitative insights into the pathogenesis and the clearance of viruses such as the human immunodeficiency virus (HIV) and influenza [6,19], and so it is a promising approach to resolving the paradox. Early modelling work assumed that measles-specific immune cells are not infected by the virus and thus do not capture the cost of immune activation post infection [10,20]. Recently, progress was made in the development of a model that fits *in vivo* data from measles infection in rhesus macaques [16]. Unlike previous models, the recent model developed in [16] includes a predatory feedback between the virus and measles-specific T cells. The simulations in [16] provide quantitative evidence suggesting that the clearance of acute viremia is dominated by T-cell immunity during measles infection, but dominated by target-cell depletion when viral fitness is increased to simulate CDV infection. This model has yet to be mathematically analysed beyond simulations and we still lack insight into the underlying causes of this paradox.

In this paper, we used control theory to investigate the feedback mechanism in the recently proposed model in [16], which helps resolve the measles paradox, see tutorial in box 2. We determined mathematical conditions for the control and robustness of viral clearance. In particular, we helped characterize the importance of immunosuppression for viral clearance as well as the switch between T-cell mediated control and lymphocyte depletion. Our results are consistent with experimental data and help to untangle complex feedback mechanisms underlying similarities and differences in the dynamics of MV and CDV in their natural hosts. Together, our results advance understanding of virus–host interactions leading to the control of acute viremia and immunosuppression during morbillivirus infections.

# 2. Background

In this section, we provide background on the experimental data and on the mathematical model from [16], which is analysed throughout the paper.

**Box 1.** Control theory finds applications in immunology.

Immunology and control theory share a common interest: the understanding of *feedback control* [18,21–24]. The immune system applies various immune responses to control viral infections [5,10,22,25]. Viral infections such as measles disturb biological feedback and impair immune reponses by destroying immune cells [10,12,18]. Virus-specific immune responses operate as a feedback control to clear viral particles and destroy infected cells [10,21,26]. Thus, the feedback between viruses and immune cells predominantly determines clinical outcomes in the absence of treatment [18,22,26,27]. Indeed, it is important to understand how immunological feedback controls viral infections.

With this in mind, control theory is applied to elucidate the fundamental principles and paradigms which underpin immunological feedback. Control theory is used as a modelling and simulation tool to monitor and predict the performance of immune responses against viral infections [26–28]. Here, control theory is used to explain different outcomes observed during acute morbillivirus infections. Analysis is completed for a particular mathematical form of feedback, sliding mode control [23,24,26], which is consistent with viral clearance [27,29]. The control analysis delivers fundamental mathematical conditions to exhibit viral clearance and immunosuppression (lymphopenia) during acute morbillivirus infections. This helps to resolve issues regarding the drivers of viral clearance by tracking the contribution of feedback from virus pathogenesis and T cell responses. Moreover, applying control theory in immunology provides a theoretical framework to determine the success or failure of immune responses in the presence of virus-induced perturbations.

## 2.1. Experimental data

We studied the dynamics of acute infections by MV and CDV in their natural hosts. In particular, we considered the experimental data in [10,16], in which seven juvenile rhesus macaques were infected intratracheally with the wild-type Bilthoven strain of measles. Rhesus macaques tend to experience an increase in total lymphocyte counts, measured in the peripheral blood at 3 days post infection. Subsequently, the total lymphocyte counts decline dramatically up to day 10 post infection. Simultaneously, infectious measles viral loads increase and peak on day 10. After day 10, infectious measles viral loads decline to low levels, while the total lymphocyte counts recover. The experiments measured MV-specific T-cell response using interferon-$\gamma$ spot forming cells. The number of MV-specific T cells increases significantly from day 10, and peaks around day 14 post infection. MV-infected macaques also exhibited the typical maculopapular rash, which resolved around day 14 post infection. Thus, rhesus macaques exhibited acute measles infection as observed in humans [10].

## 2.2. Mathematical model

The basis of the current study is the mathematical model of acute measles infection constructed in [16]. The population of lymphocytes includes virus-specific activated T cells, $A$, and other susceptible lymphocytes, $S$, which include naive and memory T cells and B lymphocytes. Both $A$ and $S$ are infected by the virus, $V$, to become virus-infected lymphocytes, $I$. The model is illustrated in figure 1 and the dynamical equations are

$$\frac{\mathrm{d}A}{\mathrm{d}t} = -\beta AV + qf(V)A - (1 - f(V))(d + r)A, \tag{2.1}$$

$$\frac{\mathrm{d}S}{\mathrm{d}t} = -\beta SV + q_s\omega(t)S + r(1 - f(V))A, \tag{2.2}$$

$$\frac{\mathrm{d}I}{\mathrm{d}t} = \beta(A + S)V - \delta I - u_c(I, A), \tag{2.3}$$

$$\frac{\mathrm{d}V}{\mathrm{d}t} = pI - cV, \tag{2.4}$$

where

$$\omega(t) = \begin{cases} 1 & \text{if } t < t_d \\ 0 & \text{if } t \geq t_d \end{cases}$$

and

$$f(V) = \frac{V}{s + V}. \tag{2.5}$$

**Box 2.** Tutorial on sliding mode control theory to interpret measles and T-cell response dynamics.

Control theory is an area of research that contains powerful mathematical tools for studying models with feedback, with applications ranging from technology to biology [24,26,30–35]. In particular, control theory is a useful tool for analysis of the recent measles model from [16], which incorporates complex feedbacks between virus and the immune system. The immune system appears to apply different types of immune responses at different time points to control measles infection, which is similar to a switched control system [10,18].

Recent control analysis of viral infections in [26,28,36] applied the **reachability paradigm**, which determines the ability of a (feedback) control input to steer the temporal dynamics of a system from an initial state to a desired state in finite time [23,24]. These interdisciplinary studies revealed synergies between the specific immune response and switched control systems. A switched control system is a system in which the feedback control changes dynamically to function effectively. Switched control systems can be designed (e.g. in technology) using the *reachability paradigm* to exhibit a *sliding mode*, where the trajectories of the system are confined to a desired manifold. The design formulates a switching function $s_0(t)$ such that the desired behaviours are exhibited when the trajectories of the system exhibit an ideal sliding mode at $s_0(t) = 0$. This ideal sliding mode is important to ensure that the desired dynamics are robust to uncertainties and perturbations in the input channel [23,24,32]. The **reachability paradigm**

$$s_0 \frac{\mathrm{d}s_0}{\mathrm{d}t} < 0 \tag{3.1}$$

is a sufficient condition to guarantee that the sliding mode exists and is attainable in finite time [23,24,32]. Thus, the switched control action is designed to satisfy the reachability condition to enforce desired stable and robust dynamical behaviours [23,24,32]. Applying the reachability paradigm also delivered nonlinear and sufficient conditions for the control and robustness of the containment of HIV infection by the HIV-specific CD8+ T-cell response and antiretroviral treatment in [27,29].

In this paper, we studied virus–host interactions for MV and CDV during acute infection using the model from [16]. We implement a novel method by casting the immune control problem as a sliding-mode control problem. Thus, in the context of this paper, the reachability condition is a nonlinear and sufficient condition to enforce viral clearance by reducing the population of infected lymphocytes continuously. Unlike typical control studies, we do not design or implement any controller. The control analysis aims to provide important insights into the feedback mechanisms underpinning immunosuppression and viral clearance during morbillivirus infections.

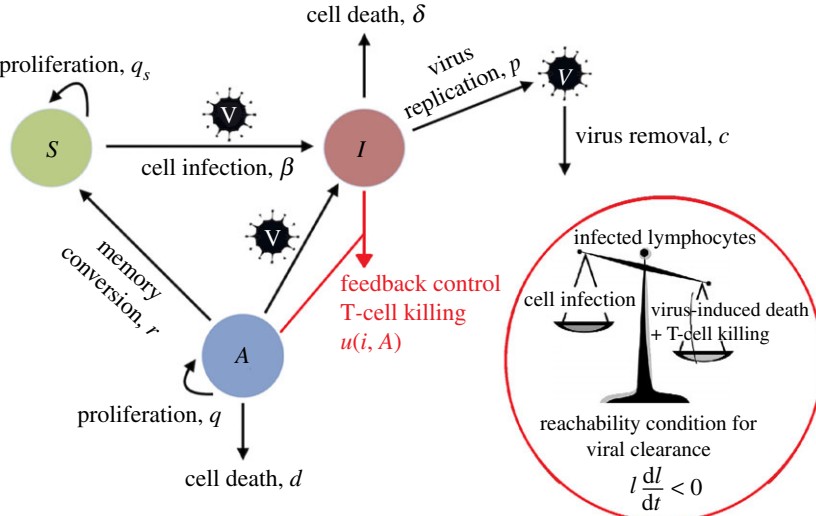

**Figure 1.** Mathematical modelling and feedback control anslysis. $S$, Susceptible lymphocytes; $A$, Activated virus-specific T cells; $I$, Infected lymphocytes and $V$, virus. Image adapted from [16].

**Table 1.** Summary of the components of the model (2.1)–(2.4).

| name | unit | symbol |
|---|---|---|
| susceptible lymphocytes | cells $\mu l^{-1}$ | $S$ |
| infected lymphocytes | cells $\mu l^{-1}$ | $I$ |
| virus-specific activated T cells | cells $\mu l^{-1}$ | $A$ |
| infectious viral load | log $TCID_{50}/10^6$ PBMC | $V$ |
| total lymphocyte count | cells $\mu l^{-1}$ | $L = S + I + A$ |
| infection rate | (log $TCID_{50}/10^6$ PBMC)$^{-1}$ day$^{-1}$ | $\beta$ |
| general lymphocyte proliferation rate | day$^{-1}$ | $q_s$ |
| general lymphocyte proliferation function | dimensionless | $\omega(t)$ |
| duration of general lymphocyte proliferation | day | $t_d$ |
| proliferation function of activated T cells | dimensionless | $f(V)$ |
| saturation constant | log $TCID_{50}/10^6$ PBMC | $s$ |
| conversion rate of activated T cells into memory T cells | day$^{-1}$ | $r$ |
| death rate of infected lymphocytes | day$^{-1}$ | $\delta$ |
| cytolytic killing function | | $u_c(I, A)$ |
| killing rate of activated T cells | (cells $\mu l^{-1}$)$^{-1}$ day$^{-1}$ | $k$ |
| proliferation rate of activated T cells | day$^{-1}$ | $q$ |
| death rate of activated T cells | day$^{-1}$ | $d$ |
| viral replication rate | $\dfrac{\log TCID_{50}/10^6\ PBMC}{(\text{cells } \mu L^{-1})\text{day}}$ | $p$ |
| viral clearance rate | day$^{-1}$ | $c$ |

Note: TCID, tissue culture infective dose; PBMC, peripheral blood mononuclear cell.

In (2.1), the first term is the mass-action rate that virus-specific activated T cells, $A$, get infected by measles virus, $V$, with infection coefficient $\beta$. In the second and third terms, some proportion $f(V)$ of activated T cells proliferate at rate $q$, while the remaining proportion $1 - f(V)$ die at rate $d$ or differentiate into memory T cells at rate $r$. The saturating function $f(V)$ models a smooth transition from greater T-cell proliferation at high viral loads to greater memory-cell differentiation and death when viral loads decline.

In (2.2), the first term is the infection rate of susceptible lymphocytes, $S$. The second term describes the observed proliferation of susceptible lymphocytes a few days post infection using the function $\omega(t)$ [16]. Susceptible lymphocytes proliferate at a rate $q_s$ during the first $t_d$ days when the proliferation function $\omega(t) = 1$ and $t < t_d$. The third term is the rate that virus-specific activated T cells, $A$, differentiate into memory T cells and join the $S$ population.

In (2.3), the first term is the rate that virus-specific activated T cells and other susceptible lymphocytes get infected and enter the infected lymphocyte population, $I$. The second term is the rate that infected lymphocytes die at rate $\delta$ due to infection. The third term is the rate virus-specific activated T cells kill infected lymphocytes. The killing process is described by the function $u_c(I, A)$, which depends on the number of infected lymphocytes and virus-specific activated T cells.

In (2.4), the first term models infected lymphocytes producing the replicated virus, $V$, at rate $p$. The second term models the removal of virus through decay or immune clearance at rate $c$.

We consider the killing process of virus-specific activated T cells as an intrinsic feedback control. We assumed that virus-specific activated T cells kill infected lymphocytes via cytolytic mechanisms which follow mass-action kinetics as in [16],

$$u_c(I, A) = kIA, \tag{2.6}$$

where $k$ is the mass-action killing coefficient.

We summarized the components of the model (2.1)–(2.4) in table 1.

The model (2.1)–(2.4) fits well with the time course of infectious measles viral loads and lymphocyte counts, measured in MV-infected rhesus macaques from [10,16]. In the studies presented below, we performed our simulations using the best-fitting parameters from [16], see tables 2 and 3.

**Table 2.** Initial conditions of the model (2.1)–(2.4) from [10,16] for measles and CDV during acute infection.

| macaques | initial conditions | | | |
| | $S_0$ | $I_0$ | $A_0$ | $V_0$ |
| --- | --- | --- | --- | --- |
| 15U | 3828 | 0 | 36.5 | $1.1 \times 10^{-5}$ |
| 46U | 3905 | 0 | 1.1 | $2.5 \times 10^{-5}$ |
| 55U | 2456 | 0 | 1.2 | $1.0 \times 10^{-5}$ |
| 67U | 4053 | 0 | 74.7 | $1.0 \times 10^{-6}$ |
| 40V | 5487 | 0 | 2.6 | $9.9 \times 10^{-6}$ |
| 43V | 3342 | 0 | 67.4 | $3.0 \times 10^{-6}$ |
| 55V | 3876 | 0 | 3.8 | $1.0 \times 10^{-4}$ |

**Table 3.** Parameter estimates of the model (2.1)–(2.4) from [10,16] for measles and CDV during acute infection.

| macaques | estimates | | | | | | | | | | |
| | $q_s$ | $t_d$ | $\beta$ | $\delta$ | $k$ | $q$ | $s$ | $d$ | $r$ | $p^*$ | $c$ |
| --- | --- | --- | --- | --- | --- | --- | --- | --- | --- | --- | --- |
| 15U | 0.075 | 5.7 | 0.143 | 0.5 | 0.005 | 0.99 | 0.0104 | 0.025 | 0.056 | 0.019 | 3 |
| 46U | 0.028 | 2.6 | 0.165 | 0.5 | 0.024 | 1.11 | 0.0006 | 0.025 | 0.016 | 0.022 | 3 |
| 55U | 0.006 | 3.2 | 0.04 | 0.5 | 0.022 | 1.67 | 0.0993 | 0.025 | 0.024 | 0.126 | 3 |
| 67U | 0.049 | 4.0 | 0.014 | 0.5 | 0.023 | 0.38 | 0.0007 | 0.025 | 0.058 | 0.414 | 3 |
| 40V | 0.006 | 7.0 | 0.013 | 0.5 | 0.017 | 1.11 | 0.0232 | 0.025 | 0.159 | 0.152 | 3 |
| 43V | 0.188 | 4.2 | 0.035 | 0.5 | 0.016 | 0.98 | 0.0997 | 0.025 | 0.119 | 0.089 | 3 |
| 55V | 0.007 | 7.0 | 0.056 | 0.5 | 0.011 | 0.59 | 0.0001 | 0.025 | 0.089 | 0.052 | 3 |

*the viral replication rate, $p$, is multiplied by 2 for CDV infection.

**Table 4.** Colour code to identify parameter sets.

| ID | colour | |
| --- | --- | --- |
| 15U | blue | ▬▬▬ |
| 46U | green | ▬▬▬ |
| 55U | red | ▬▬▬ |
| 67U | cyan | ▬▬▬ |
| 40V | magenta | ▬▬▬ |
| 43V | yellow | ▬▬▬ |
| 55V | black | ▬▬▬ |

We also used the model (2.1)–(2.4) to simulate acute CDV infections in their natural hosts, as suggested in [16]. Since the peak viral load of CDV tends to occur before the peak viral load of MV in infected macaques [9], acute CDV infection is simulated by doubling the viral replication rate, $p$. The viral fitness of MV is assumed to be lower than the viral fitness of CDV [16].

We used the colour code in table 4 to distinguish the simulations generated by each parameter set. Moreover, we restricted our simulations and analysis to the first 25 days post infection to focus on the acute phases of MV and CDV infections.

# 3. Results

## 3.1. Immunosuppression enforces viral clearance

We first studied the clearance of MV and CDV during acute infection. We analysed the mathematical model (2.1)–(2.4) from [16] to formally study the control of acute viremia using control theory, see appendix A. We found the following *reachability condition* (also see boxes 1 and 2):

$$\beta(S + A)V - \delta I - u_C(I, A) < 0, \tag{3.2}$$

which defines a dynamical condition to reduce the population of infected lymphocytes to zero, i.e. $I = 0$ and to enforce viral clearance because when $I = 0$ in (2.1)–(2.4), the viral load $V$ declines at a rate $c$.

In simulations, we can see that when the reachability condition (3.2) is not satisfied, i.e. positive, the number of infected lymphocytes and the viral loads increase, see figure 2*A*1–3 and *B*1–3. Furthermore, when the reachability condition (3.2) becomes satisfied, i.e. negative during infection, the number of infected lymphocytes declines and this imposes that the viral loads decline, see figure 2*A*1–3 and *B*1–3. These results support the proposed reachability paradigm for viral clearance and suggest that immunosuppression enforces viral clearance; virus-induced cell death, $\delta I$, and T-cell killing, $u_C(I, A)$, both contribute to viral clearance by reducing the number of infected lymphocytes. These findings support current knowledge in the field and reject the hypothesis that either mechanism solely contributes to viral clearance.

We next examined the primary mechanism controlling viral clearance. We used the reachability condition (3.2) to formulate a sufficient condition for T-cell killing to enforce viral clearance,

$$\beta(S + A)V < u_C(I, A). \tag{3.3}$$

This condition reflects the balance between the production, $\beta(S + A)V$, and killing, $u_C(I, A)$, of infected lymphocytes. Consequently, we examined the reachability condition (3.3) during the progression of acute measles infection. The killing of infected lymphocytes is lower than the production of infected lymphocytes during the early days post measles infection, see figure 2*A*4. Afterwards, the killing of infected lymphocytes exceeds the production of infected lymphocytes during acute measles infection, see figure 2*A*4. Thus, the killing of infected lymphocytes becomes sufficient to satisfy the reachability condition (3.3) for the clearance of acute viremia. These results are consistent with previous findings which suggest that T-cell immunity controls acute measles infection [10,12,16,18].

Furthermore, we investigated how the control of acute viremia relates to the skin rash observed during measles infection because the measles skin rash is indicative of CD4+ and CD8+ T-cell responses against measles [10,12,18]. The reachability condition for viral clearance (3.2) tends to be satisfied before the onset of the rash and infectious viral loads start declining before or during the rash to undetectable levels, see figure 2*A*1–*A*3. The killing of infected lymphocytes also tends to exceed the production of infected lymphocytes before the onset of the rash, see figure 2*A*4. These results suggest that the measles rash is a clinical sign for the successful control of acute measles infection by T-cell immunity in the peripheral blood. Thus, these results are in line with experimental findings in [10,16,18].

In addition, we determined the primary mechanism for viral clearance during acute CDV using the condition (3.3). The killing of infected lymphocytes remains lower than the production of infected lymphocytes during acute CDV infection, see figure 2*B*4. These results support previous findings which suggest that the depletion of lymphocytes by CDV enforces the clearance of acute CDV infection [13–16]. Together, these results indicate that the balance between the production and killing of infected lymphocytes determines the control of acute viremia by virus-induced lymphocyte death or by T-cell immunity. Thus, the proposed control theory methods provide an analytical framework to gain insights into the switch from lymphocyte depletion-mediated control to immune-mediated control, and more broadly for the control mechanisms of viral clearance across the morbilliviruses.

## 3.2. The control of acute viremia causes the reduction of the total lymphocyte count

Since measles and CDV preferentially infect lymphocytes and replicate in lymphoid tissues [7–9], the literature hypothesized that the total lymphocyte count declines due to virus-induced lymphocyte death. To advance understanding of the mechanisms of immunosuppression, we investigated the reduction of the total lymphocyte count. The model (2.1)–(2.4) defines the total lymphocyte count as

$$L = S + I + A \tag{3.4}$$

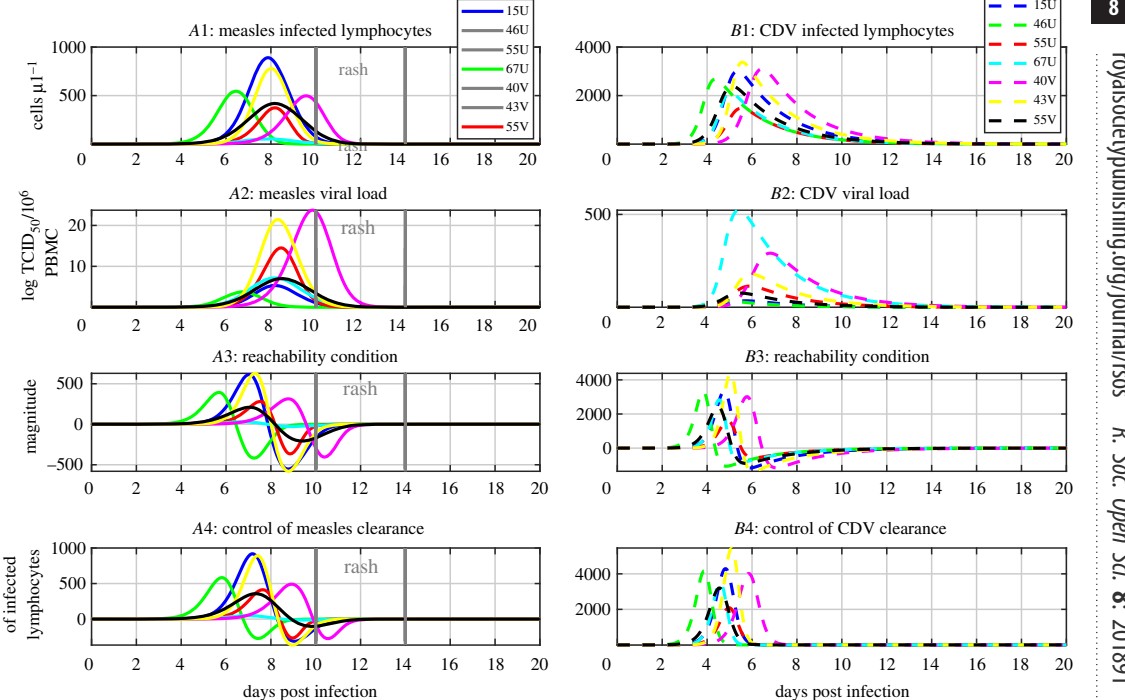

**Figure 2.** Immunosuppression enforces viral clearance. The left panels and solid lines are related to acute measles infection. The right panels and dashed lines relate to acute CDV infection. Panels *A*1 and *B*1 show the time course of the population of infected lymphocytes. Panels *A*2 and *B*2 show the time course of infectious viral load, *V*, from simulations of the model (2.1)–(2.4). Panels *A*3 and *B*3 show the time course of the reachability condition (3.2). Panel *A*4 and *B*4 show the time course of (3.3) and the difference between the magnitude of the production and killing of infected lymphocytes. The different colours represent different parameter sets, see tables 3 and 4.

and we differentiated (3.4) over time to obtain the differential equation representing the variation over time of the total lymphocyte count,

$$\frac{\mathrm{d}L}{\mathrm{d}t} = q_s \theta(t)S + qf(V)A - \delta I - u_c(I, A) - (1 - f(V))(d)A. \tag{3.5}$$

We observed that the term $(1 - f(V))(d)A$ is negligible when the T-cell response is activated, i.e. $f(V) \approx 1$. The expression (3.5) reflects that viral infection reduces the total lymphocyte count by destroying infected lymphocytes, $\delta I$ and highlights that T-cell immunity also contributes to the reduction of the total lymphocyte count by killing infected lymphocytes, $u_c(I, A)$. This rejects the hypothesis that the total lymphocyte count declines only due to virus-induced lymphocyte death. Together, the analytical results (3.2) and (3.5) suggest that immunosuppression and viral clearance are enforced by the same mechanism, i.e. the death of infected lymphocytes due to the virus and T-cell immunity. These results have not been reported previously.

We also studied the relationship between viral control and immunosuppression in order to elucidate the measles paradox. We examined the control of viral clearance (3.3) along the time course of the total lymphocyte count (3.4) during acute measles and CDV infection. The killing of infected lymphocytes exceeds the production of infected lymphocytes while the total lymphocyte count declines during acute measles infection, suggesting that T-cell immunity enforces measles viral clearance during the typical drop of the total lymphocyte count in acute MV infections, see figures 2*A*3 and 3*A*1. This has not been found by previous studies. Furthermore, the killing of infected lymphocytes remains below the production of infected lymphocytes while the total lymphocyte count declines during acute CDV infection, see figures 2*B*3 and 3*B*1. These results suggest that CDV impairs the immune system even after viral clearance. This is in line with previous findings because CDV infection is often fatal to their natural hosts [13–16].

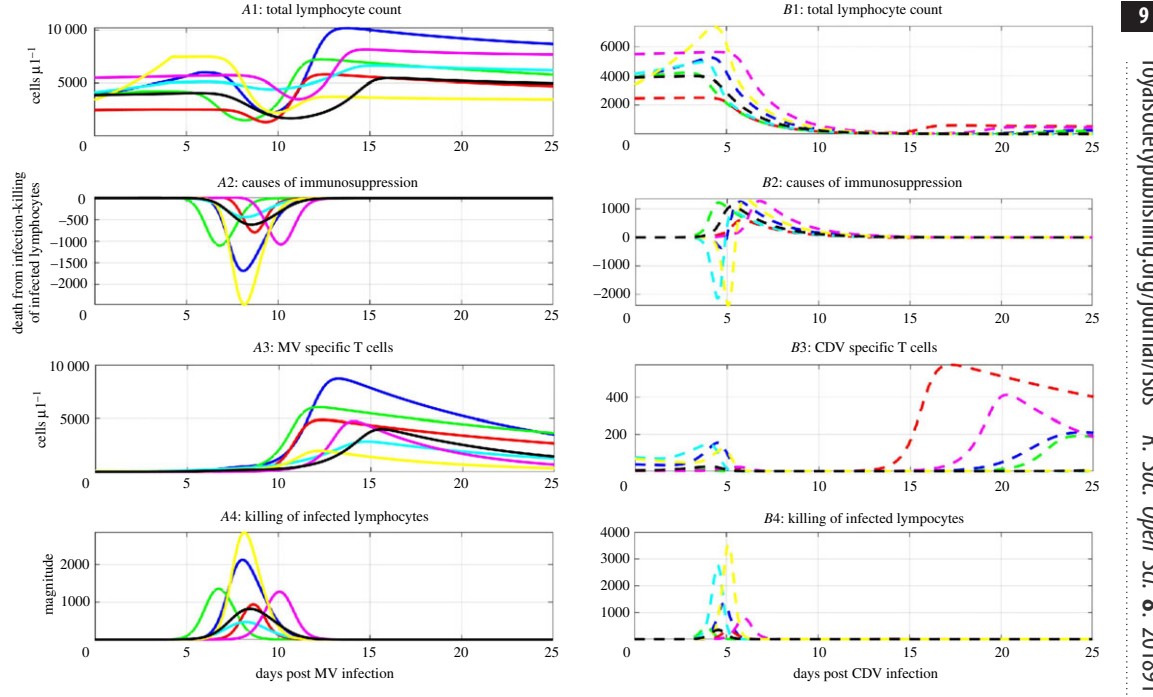

**Figure 3.** The control of acute viremia causes the reduction of the total lymphocyte count. The left panels and solid lines are related to acute measles infection. The right panels and dashed lines are related to acute CDV infection. Panels $A1$ and $B1$ show the time course of the total lymphocyte count, $L = S + I + A$ from the model (2.1)–(2.4). Panels $A2$ and $B2$ show the time course of the difference between the virus-induced death of infected lymphocytes and the killing of infected lymphocytes using (3.6). Panels $A3$ and $B3$ show the time course of the population of virus-specific activated T cells, $A$. Panels $A4$ and $B4$ show the time course of the magnitude of the killing of infected lymphocytes (2.6). The different colours represent different parameter sets, see tables 3 and 4.

Next, we investigated the primary mechanism contributing to immunosuppression. We considered the function

$$D_L = \delta I - u_c(I, A) \tag{3.6}$$

in order to compare the impact of virus-induced immunosuppression and T-cell killing on the reduction of the total lymphocyte count during acute measles and CDV infection. The loss of lymphocytes due to viral infection becomes lower than the T-cell killing of lymphocytes during acute measles infection, suggesting that T-cell immunity becomes the main cause of the typical drop of the total lymphocyte count during acute MV infection, see figure $3A2$. This rejects the hypothesis that the total lymphocyte count predominantly declines due to MV-induced lymphocyte death [3,10,16], and contradicts the notion that virus-induced lymphocyte death mainly reduces the total lymphocyte count during typical cases of measles infection. By contrast, the loss of lymphocytes due to viral infection tends to be higher than the T-cell killing of lymphocytes during acute CDV infection, see figure $3B2$. These results are consistent with previous studies [13,16] and suggest that the total lymphocyte count mainly declines due to the virus-induced cell damage during acute CDV infection, see figure $3B1,2$.

Furthermore, we examined the time course of the killing action and virus-specific activated T cells during acute measles infection. The killing action against measles reaches its maximal value a few days post infection, before the peak of viremia, then declines due to viral clearance, despite subsequent increases in the number of measles-specific T cells, see figures $2A2$ and $3A3,4$. This has not been previously described in [10,16]. These results suggest that though T-cell killing reduces the total lymphocyte count to control acute viremia, T-cell killing does not impair the recovery of the total lymphocyte count. This is consistent with previous findings on MV infection [10,12,16,18]. The MV-specific T-cell response thus exhibits some robustness to measles virus predation. Since this robustness property is not described in the literature, it warrants further investigation.

We also examined the time course of the killing action and virus-specific activated T cells during acute CDV infection. Unlike in MV infection, the number and the killing action of CDV-specific activated T cells

tend to peak simultaneously around the peak of viremia on day 5 post CDV infection, see figure 3B3 and B4. These dynamics have not been described in the literature, in part due to the challenge in measuring these dynamics. The number of CDV-specific T cells tends to be smaller than the number of MV-specific T cells due to extensive predation of CDV, see figure 3A3,4 and B3,4. Surprisingly, in few cases, the killing against CDV is higher than the ones observed during MV infection, supporting the notion that T-cell immunity may mount a strong response against CDV, but which is still insufficient to control CDV infection. The killing action remains almost null after the peak, since the virus is cleared. These results agree with previous studies which suggest that CDV infection impairs the immune system [13,16]. Though the number of CDV-specific T cells rebounds in our simulations in figure 3B3,4, this may not be observed *in vivo*, since CDV infection tends to be fatal [13].

## 3.3. T-cell immunity ensures the robustness of measles viral clearance

We next studied the robustness properties of viral clearance during acute measles infection. We used the reachability paradigm to hypothesize that T-cell immunity ensures the robustness of viral clearance as long as the killing of infected lymphocytes exceeds the production of infected lymphocytes (3.3). As in [12,16,18], we introduced a perturbation that induces the loss of activated virus-specific T cells (A). The loss of activated cells reflects a perturbation that the system may encounter in nature as activated cells are destroyed by the predation of MV and CDV, concurrent immunosuppressive infections such as HIV/AIDS or concurrent immunosupressive treatments such as the ones used for leukaemia and transplantation [3,37]. We tested whether viral clearance continues or stops in the presence of a reduced number of measles-specific T cells after the peak of acute viremia, see appendix A. Measles viral clearance continues despite the loss of a large number of measles-specific T cells, see figure 4A1,A2. This is paradoxical, since MV-specific T cells are thought to play a major role during viral clearance [10,12,16,18]. As this reduction in the number of measles-specific T cells does not cause the reachability condition to fail (3.3), see figure 4A3, the former observation is expected from a control engineering standpoint; it is in good agreement with the robustness properties of sliding mode control [24]. From a control engineering viewpoint, this result suggests that T-cell immunity ensures that the clearance of acute measles infection exhibits some robustness to virus predation and changes in the number of measles-specific T cells. This is consistent with previous studies which show that T-cell immunity exhibits robustness properties [26,27].

We also assessed the impact of the reduction in the number of susceptible lymphocytes on viral clearance. We considered the case in which susceptible lymphocytes $S$ do not proliferate during measles infection. As expected, the number of susceptible lymphocytes is reduced in the absence of proliferation during the first days post infection, see figure 4A4. This reduction decreases viral loads and affects the dynamics of viral clearance, see figure 4A5,6. This result is paradoxical, since target cell availability is thought to play a minor role during acute measles infection [16,18]. From the sliding mode control perspective, these observations are expected because this perturbation takes place before the reachability condition is satisfied, see figure 4A6. This suggests that viral dynamics are sensitive to biological perturbations when T-cell killing is not yet sufficient to drive viral clearance. This is consistent with experimental findings in [18]. Overall, these results suggest that the balance between the production and killing of infected lymphocytes also determines when T-cell immunity ensures the robustness of viral clearance. These findings are consistent with the dynamics of acute MV infection. Interestingly, these robustness properties motivate investigations for early therapeutic interventions, which induce immunosuppresion before the onset of the rash, to yield desirable clinical outcomes.

## 3.4. The control of viral clearance may dynamically switch from T-cell immunity to the depletion of lymphocytes

We finally studied the dynamical control of measles infection in the presence of a complete and rapid suppression of measles-specific activated T cells after the peak of acute viremia, see appendix A. The results show that after the depletion of measles-specific T cells, measles viral loads first rebound and then decline, see figure 5A1. The condition (3.3) fails after the depletion of measles-specific T cells, indicating that this killing of infected lymphocytes falls below the production of infected lymphocytes, as shown in figure 5A2. This observation is consistent with the failure to maintain the desired behaviour prescribed by the ideal sliding mode, when the controller fails to maintain the reachability

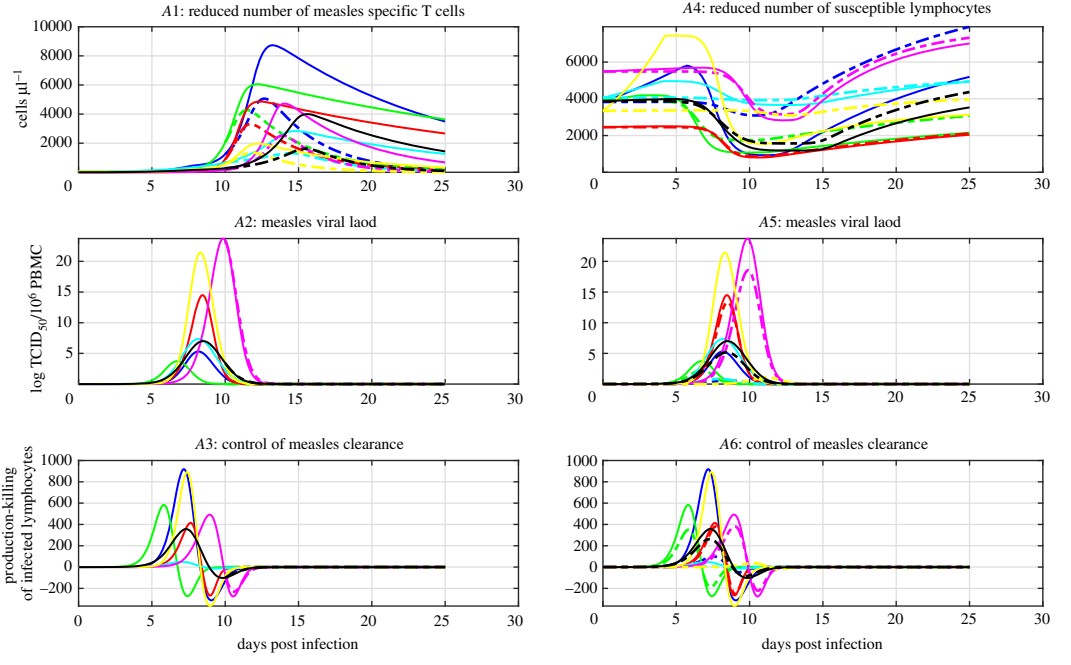

**Figure 4.** T-cell immunity ensures the robustness of measles viral clearance. Panels A1–3 are dedicated to the experiments in which the number of measles-specific T cells are reduced after day 10 post infection. Panels A4–6 are dedicated to the experiments in which susceptible lymphocytes S do not proliferate. The solid lines refer to the natural course of the infection and the dashed lines refer to the simulation experiment. We used the colour code in table 4 to identify each macaque. Panel A1 shows the time course of the population of virus-specific activated T cells, A. Panel A4 shows the time course of the total lymphocyte count, L. Panels A2 and A5 show the time course of infectious viral load, V. Panels A3 and A6 show the time course of (3.3), the difference between the magnitude of the production and killing of infected lymphocytes. The different colours represent different parameter sets, see tables 3 and 4.

condition in the presence of perturbations [24]. Together, these results suggest that the control of viral clearance dynamically switches from T-cell immunity to the depletion of lymphocytes, when the measles-specific T-cell response is impaired. This is consistent with severe or fatal cases of measles infection in individuals who are immunosuppressed, such as children with HIV or children treated for leukaemia [3,37]. Thus, the proposed analytical methodology could enable disease modellers to elucidate the control and robustness of viral clearance across morbilliviruses during the different phases of infection.

# 4. Discussion

The control analysis of MV and CDV infections contributes to the resolution of the measles paradox. Our control analysis builds on dynamical equations and experiments, which describe well acute MV and CDV infections in their natural hosts [10,13,16,18]. We applied the nonlinear framework of sliding mode control theory, which enabled a better understanding of modelling and parameter uncertainties that have a bearing on the control of acute viremia [26,27]. Studying viral control and temporal dynamics using reachability analysis contrasts with steady-state analysis, which is more common in the literature [16,26,27,38,39]. This reachability analysis delivers the reachability condition (3.2), a nonlinear condition that tracks the requirements for viral clearance during the progression of the infection. The time-varying nature of the reachability condition (3.2) supports the notion that immunological requirements to enforce viral clearance vary during the progression of the infection. This is consistent with findings on COVID-19, HIV and LCMV infection dynamics [6,26,27,40].

Our control analysis delivers an analytical framework to determine virus–host interactions leading to immunosuppression and viral clearance during morbillivirus infections. Although early work in [41] suggested that immunosuppression is due to apoptosis of uninfected cells, our findings are consistent with recent experimental findings in [8], which support the explanation that lymphocyte counts decline because lymphocytes infected by measles die. Furthermore, our findings show that immunosuppression enforces viral clearance as the continuous decline of the population of infected lymphocytes enforces the

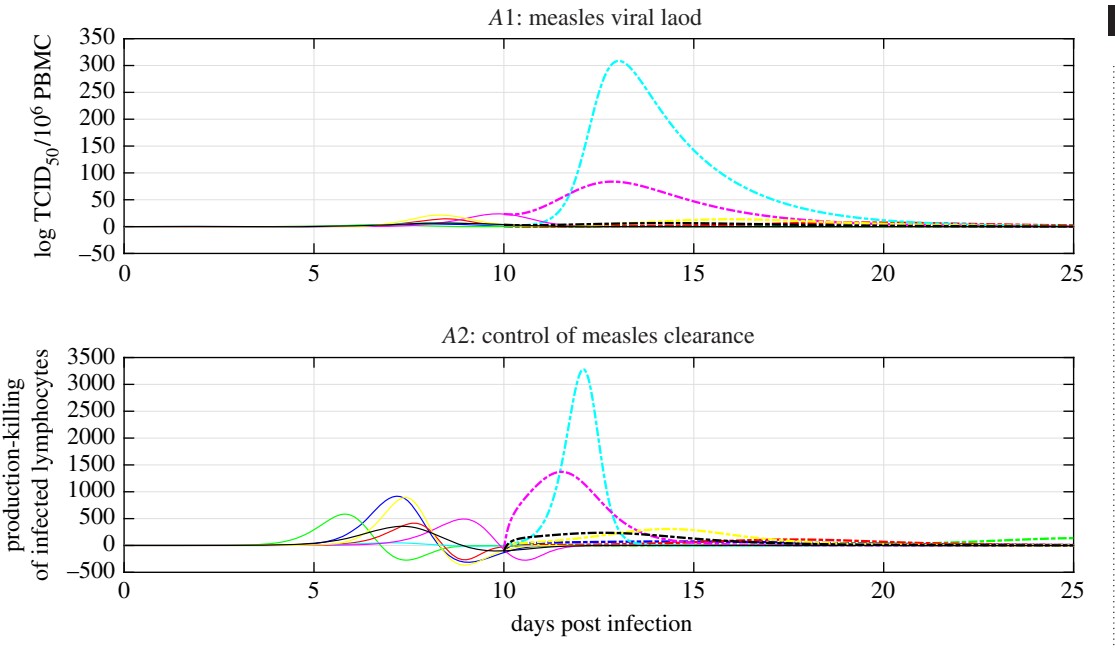

**Figure 5.** The control of viral clearance may dynamically switch from T-cell immunity to the depletion of lymphocytes. Panels show the results of the simulations in which measles-specific T cells are depleted after day 10 post infection. The solid lines refer to the natural course of the infection and the dashed lines refer to the simulation experiment. We used the colour code in table 4 to identify each macaque. Panel $A1$ shows the time course of infectious viral load, $V$. Panel $A2$ shows the time course of (3.3), the difference between the magnitude of the production and killing of infected lymphocytes.

reduction of infectious viral loads. This is supported by experimental data, noting the observations in [12,18] that viral clearance starts when productively infected cells per million peripheral blood mononuclear cells (PBMC) decline continuously. As infected lymphocytes predominately die due to viral infection or T-cell killing, both processes contribute to immunosuppression and viral clearance. Thus the process, which predominantly reduces the number of infected lymphocytes, predominantly contributes to immunosuppression and viral clearance during morbillivirus infections. Our findings could be validated experimentally by investigating the death of B and T lymphocytes, as a large number of B and T lymphocytes tend to be infected and destroyed during measles and CDV infections [8,9,12,14,15].

Previous studies found that the onset of the typical measles rash coincides with the appearance of virus-specific neutralizing antibodies and T lymphocytes, which correlate with a rapid decrease in viral load [8,10,18]. Our findings add that the control of measles viral clearance tends to be effective in the peripheral blood before or on the onset of the rash because the population of measles-infected lymphocytes tends to decline before or during the onset of the measles rash. Our findings are consistent with experimental observations in [8,18], as a maculopapular abdominal skin rash appears between days 10 and 14 post measles infection. In addition, the number of productively infected cells PBMC and MV viral load in PBMC decline before or on the onset of this typical measles rash. Since T-cell immunity controls measles infection in the blood, T-cell immunity might also control measles infection in tissues, since (i) infiltration of CD4+ T cells and CD8+ T cells have been observed in the skin layers [42,43], (ii) MV-infected cells tend to decrease to undetectable levels in the skin by the end of the rash [8], and (iii) macaques depleted of CD8+ T cells tend to have a more severe and prolonged rash [12,18]. Since infectious viral loads are not detectable after the rash in the peripheral blood, but remain detectable in tissues [8,10], T-cell killing appears to enforce viral clearance in the peripheral blood, and then in the tissues. The typical measles rash becomes a clinical sign indicating that measles-specific T cells successfully control infectious viral load.

The balance between the production and killing of infected lymphocytes highlights that T-cell killing becomes sufficiently large to enforce viral clearance on its own by producing a robust feedback control, which overcomes the predatory feedback of the virus on lymphocytes. Thus, measles-specific T-cell killing becomes the primary mechanism for the death of infected lymphocytes. Since B cells have the highest number of measles-infected cells than other types of lymphocytes [8,10,16,18], our findings suggest that measles-specific T-cell killing becomes the primary mechanism causing impaired or

delayed B-cell responses to enforce clearance of infectious viral loads. Our results also suggest that measles-specific T-cell killing becomes the primary mechanism for the loss of immunological memory, since memory T cells have the highest number of measles infection than other T cells [8]. In contrast with early work in [41], our results suggest that measles-specific T-cell killing becomes the primary mechanism that renders infected individuals temporarily vulnerable to opportunistic infection. Since B and T cells are also the main targets of other morbilliviruses [7–9], our findings may also apply when the specific T-cell response controls these infections. The dominant role of measles-specific T-cell killing during immunosuppression and viral clearance may contribute to low mortality rates reported for measles, as compared with high fatality rates observed in CDV [9].

When virus-specific T cells kill infected lymphocytes during measles and CDV infection, the feedback control depicts a time-varying killing action having a bell shape. From a control standpoint, this bell shape can be interpreted as a continuous switch control signal. Such switched control signals are useful to overcome uncertainties and perturbations in the system, and to avoid undesirable impacts related to prolonged activation. In the context of measles infection, T-cell killing increases to achieve viral clearance and then decreases to allow recovery of the total lymphocyte count, while ensuring some robustness to changes in the number of virus-specific T cells during viral clearance. Killing outputs during measles infection tend to be higher than killing output during CDV infection. This supports experimental findings suggesting that CDV impairs the killing action of CDV-specific T-cell responses [9,14,15].

Our control analysis allows the robustness of viral clearance to be explained analytically using the predictive capacity of the reachability analysis. The reader may recall that sliding mode dynamics exhibit robustness when the reachability condition is satisfied [23,24,32]. As the proposed reachability condition suggests the reduction of the number of infected lymphocytes, it is expected that when the population of infected lymphocytes declines continuously, viral clearance exhibits some robustness properties. Thus, the decline of infectious viral loads exhibits some robustness to biological perturbations, such as the reduction in the number of virus-specific T cells, or the addition of susceptible lymphocytes, as observed in [16]. Viral dynamics are expected to be perturbed by the reduction of the number of CD8+ T cells and virus-specific T cells, before the decline of the population of infected lymphocytes. Hence, our results provide a framework to understand why measles viral clearance continues or fails in the experiments in [12,16,18].

Our findings reinforce that it is important to advance understanding of what constitutes protective and pathogenic aspects of immune responses to viral infections [22]. Acute measles infection differs from other viral infections including COVID-19. Though immunosuppression is observed during measles and COVID-19, immunosuppression is caused by different pathogenesis [22]. Unlike measles, clinical outcomes of COVID-19 may worsen following viral clearance due to immunopathology [22]. Thus, understanding the details of virus-specific immune responses is key to control viral infections such as measles and COVID-19 at the individual level and at the population level.

In summary, our study applies methods for feedback control systems to disease models, and draws broader conclusions about the conditions for the control of morbillivirus during the progression of the infection.

## Software

This work has been conducted using MATLAB R2019a.

Data accessibility. The datasets supporting this article have been uploaded as part of the electronic supplementary material.

Authors' contributions. Substantial contributions to conception and design: A.J.N.A. substantial contributions to analysis and interpretation of data: A.J.N.A., E.J.H. and S.K.S. Drafting the article or revising it critically for important intellectual content: all authors. Final approval of the version to be published; and agreement to be accountable for all aspects of the work in ensuring that questions related to the accuracy or integrity of any part of the work are appropriately investigated and resolved: all authors. A.J.N.A. conceived of the study, designed the study, conducted the analytical analysis, produced the simulations and figures, wrote the first draft of the article and the response to reviewers, and coordinated the study. E.J.H. contributed to the analysis and interpretation of results, helped with the artwork, and helped draft the manuscript and the response to reviewers. N.K. contributed to the interpretation of results, and helped draft the manuscript and the response to reviewers. P.K. helped draft the manuscript and the response to reviewers. S.S. contributed to the analysis and interpretation of results, helped draft the manuscript and the response to reviewers, and supervised the study. All authors gave final approval for publication.

Competing interests. We declare we have no competing interests.

Funding. S.K.S. and A.J.N.A gratefully acknowledge the support of the EPSRC via grant reference no. EP/J018295/1. P.K. and A.J.N.A. gratefully acknowledge the support of the Australian Research Council via grant reference no. DP180101512. The authors gratefully acknowledge the support of The University of Sydney—UCL Mobility Scheme.

Acknowledgements. E.J.H. would like to thank Judith and David Coffey for their contribution.

# Appendix A. Methods

## A.1. Analysis to determine the control of viral clearance

We investigated the control of viral clearance using the reachability paradigm from the sliding-mode control [23,24]. We assumed that the control objective of the virus-specific T-cell response in the model (2.1)–(2.4) is to enforce protective immunity by killing lymphocytes infected by the virus. We start the reachability analysis by defining the switching function

$$s_I(t) = I,$$ (A 1)

which defines a control objective for virus-specific activated T cells to reduce the population of infected cells to zero, i.e. $s_I(t) = I = 0$. The manifold $s_I(t) = I = 0$ is appropriate to exhibit viral clearance because when $I = 0$ in the model (2.1)–(2.4), the viral load $V$ declines at a rate $c$. The proposed reachability condition for viral clearance is

$$s_I \frac{\mathrm{d}s_I}{\mathrm{d}t} < 0$$
$$\Rightarrow I(\beta(S+A)V - \delta I - u_C(I, A)) < 0$$
$$\Rightarrow \beta(S+A)V - \delta I - u_C(I, A) < 0,$$ (A 2)

since $s_I(t)$ is non-negative at all times, because it relates to the number of infected cells. As soon as the left-hand side of (3.2) becomes negative, the reachability condition is satisfied, and the sliding manifold $s_I(t) = I = 0$ becomes attractive, which triggers the decline of infected cells $I$ along with the decline of the viral load. From the above reachability analysis, we found the following condition to determine the control of viral clearance:

$$\beta(S+A)V - u_C(I, A) < 0.$$ (A 3)

This condition is a sufficient condition to satisfy the reachability condition (3.2) and enforce viral clearance. The condition (A 3) represents the balance between the production of infected lymphocytes, $\beta(S+A)V$, and killing of infected lymphocytes, $u_C(I, A)$. We hypothesize based on the reachability condition (3.3) that virus-specific T-cell immunity primarily enforces viral clearance when the magnitude of the killing of infected lymphocytes becomes higher than the production of infected lymphocytes. On the other hand, when this killing action is negligible or impaired, we hypothesize that virus-induced lymphocyte death primarily enforces viral clearance by satisfying the reachability condition (3.2).

## A.2. Analysing the reduction of the total lymphocyte count during measles and canine distemper virus infection

We studied the time course of the total lymphocyte count to investigate the main reason for the loss of lymphocytes during measles and CDV infection. The total lymphocyte count is

$$L = S + I + A$$ (A 4)

and we differentiated (3.4) over time to obtain the variation over time of the total lymphocyte count:

$$\frac{\mathrm{d}L}{\mathrm{d}t} = q_s \theta(t)S + qf(V)A - \delta I - u_c(I, A) - (1 - f(V))(d)A.$$ (A 5)

We observed that the term $(1 - f(V))(d)A$ is negligible when the T-cell response is activated, i.e. $f(V) \approx 1$. Therefore, the total lymphocytes count reduces because the viral infection and T-cell killing causes the death of infected lymphocytes. To determine the main reason for immunosuppression, we compared the loss of lymphocytes induced by viral infection and the loss of lymphocytes induced by T-cell

killing by monitoring

$$D_L = \delta I - u_c(I, A) \qquad (A\,6)$$

during acute measles and CDV infection. We hypothesized that T-cell killing primarily reduces the total lymphocyte count when (3.6) is negative, and viral infection primarily reduces the total lymphocyte count when (3.6) is positive.

## A.3. Analysing the robustness of viral clearance

We analysed the robustness of viral clearance in the presence of biological uncertainties and perturbations using knowledge from the reachability paradigm. We considered the dynamics of measles infection to conduct three simulation experiments to understand the robustness of viral clearance. In the first experiment, we reduced the number of virus-specific activated T cells after day 10 post infection by inserting the term $-\gamma_d A$ where $\gamma_d = 0.1$ day$^{-1}$ in the dynamical equation of virus-specific activated T cells in (2.1). In our second experiment, we assumed that susceptible lymphocytes, $S$, do not proliferate, i.e. $q_s = 0$, during the infection. In our third experiment, we simulated a rapid suppression of virus-specific T cells from day 10 post infection using $\gamma_d = 3$ day$^{-1}$.

From sliding-mode control theory [23,24], the sliding-mode dynamics exhibit complete robustness *matched uncertainties*. Matched uncertainties are uncertainties implicit in the input channels of the control signals such as variations in parameter values, external perturbations and discrepancies between the model and the real system. When the reachability condition is satisfied, the dynamics still exhibit some robustness to uncertainties and perturbations while the trajectories are approaching the sliding manifold. When the reachability condition is not satisfied, the system is sensitive to uncertainties and perturbations. Thus, we hypothesized that T-cell immunity sustains viral clearance as long as the killing action exceeds the production of infected lymphocytes, i.e. condition (3.3) is true.

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
