## [Peer Review File · Royal Society Open Science]

Review History

RSOS-201891.R0 (Original submission)

Review form: Reviewer 1

Is the manuscript scientifically sound in its present form?

Yes

Are the interpretations and conclusions justified by the results?

Yes

Is the language acceptable?

Yes

Do you have any ethical concerns with this paper?

No

Have you any concerns about statistical analyses in this paper?

No

Recommendation?

Accept with minor revision (please list in comments)

Comments to the Author(s)

This version of the paper is much improved. The additions to the discussion on pages 6,7,8, 9 did a good job of putting the mathematical results in the context of the natural histories of the two diseases being considered, and clarified the points of agreement with existing experimental findings. The statements on page 9 "T-cell immunity enforces measles clearance during the typical drop in total lymphocyte count" and "this has not been found in previous studies" further emphasizes the new contribution of this paper. The statement on page 15 "As this reduction in the number of measles-specific T cells does not cause the reachability condition to fail..." emphasizes the interpretive value of the analysis done in this paper. I think the authors addressed my primary concern that this section was short-changed and inadequate before.

I'm not sure what I think about Box 1. I thought that the paper needed to be expanded in its tutorial either into control systems for biologists or biology for control systems. I think the above mentioned changes made this a much more interesting paper for control systems people, as it explained the biology very well. Box 1 is a good tutorial on sliding mode theory for people already familiar with control systems, I worry it may not be general enough for a mixed audience. The concept of an invariant manifold within a state space, for instance, or of a distance function $s_0(t)$, are not going to be commonly understood topics outside of the people who already understand them. I would recommend Box 1 be re-written at a higher (less detailed) level with the mathematical biology community in mind. Focus on the implications, refer to other good survey articles for the mathematical details.

Review form: Reviewer 2

Is the manuscript scientifically sound in its present form?

Yes

Are the interpretations and conclusions justified by the results?

Yes

Is the language acceptable?

Yes

Do you have any ethical concerns with this paper?

No

Have you any concerns about statistical analyses in this paper?

No

Recommendation?

Accept as is

Comments to the Author(s)

I suggest to accept the paper.

Decision letter (RSOS-201891.R0)

Dear Dr Anelone

On behalf of the Editors, we are pleased to inform you that your Manuscript RSOS-201891 "Control theory helps to resolve the measles paradox" has been accepted for publication in Royal Society Open Science subject to minor revision in accordance with the referees' reports. Please find the referees' comments along with any feedback from the Editors below my signature.

Please submit your revised manuscript and required files (see below) no later than 7 days from today's (ie 17-Feb-2021) date. Note: the ScholarOne system will 'lock' if submission of the revision is attempted 7 or more days after the deadline. If you do not think you will be able to meet this deadline please contact the editorial office immediately.

on behalf of Dr Robert MacKay (Associate Editor) and Mark Chaplain (Subject Editor)
openscience@royalsociety.org

Associate Editor Comments to Author (Dr Robert MacKay):

The paper looks good now. The only thing is that one reviewer questions how useful Box 1 will be. Perhaps it would work better to make two boxes, one for biologists to explain in simple language how you use control theory, one for control theorists to explain how you apply it to biology? The non-trivial issue for both seems to me that you are not using control theory in the sense of designing a feedback to achieve a purpose, rather you are interpreting nature's solution as implementing a control strategy.

Reviewer comments to Author:

Reviewer: 1

Comments to the Author(s)

This version of the paper is much improved. The additions to the discussion on pages 6,7,8, 9 did a good job of putting the mathematical results in the context of the natural histories of the two diseases being considered, and clarified the points of agreement with existing experimental findings. The statements on page 9 "T-cell immunity enforces measles clearance during the typical drop in total lymphocyte count" and "this has not been found in previous studies" further emphasizes the new contribution of this paper. The statement on page 15 "As this reduction in the number of measles-specific T cells does not cause the reachability condition to fail..." emphasizes the interpretive value of the analysis done in this paper. I think the authors addressed my primary concern that this section was short-changed and inadequate before.

I'm not sure what I think about Box 1. I thought that the paper needed to be expanded in its tutorial either into control systems for biologists or biology for control systems. I think the above mentioned changes made this a much more interesting paper for control systems people, as it explained the biology very well. Box 1 is a good tutorial on sliding mode theory for people already familiar with control systems, I worry it may not be general enough for a mixed audience. The concept of an invariant manifold within a state space, for instance, or of a distance function $s_0(t)$, are not going to be commonly understood topics outside of the people who already understand them. I would recommend Box 1 be re-written at a higher (less detailed) level with the mathematical biology community in mind. Focus on the implications, refer to other good survey articles for the mathematical details.

Reviewer: 2

Comments to the Author(s)

I suggest to accept the paper.

===PREPARING YOUR MANUSCRIPT===

===PREPARING YOUR REVISION IN SCHOLARONE===

Author's Response to Decision Letter for (RSOS-201891.R0)

See Appendix A.

Decision letter (RSOS-201891.R1)

Dear Dr Anelone,

It is a pleasure to accept your manuscript entitled "Control theory helps to resolve the measles paradox" in its current form for publication in Royal Society Open Science.

You have indicated that "The datasets supporting this article have been uploaded as part of the supplementary material"; however, you do not appear to have submitted any electronic supplements (ESM) with the paper. Please can you send your ESM to the editorial office at the email address below as soon as possible?

on behalf of Dr Robert MacKay (Associate Editor) and Mark Chaplain (Subject Editor)
openscience@royalsociety.org

Associate Editor Comments to Author (Dr Robert MacKay):
Associate Editor

Comments to the Author:

~The revised paper looks excellent. I strongly recommend publication.

Appendix A

Manuscript Ref. No.: RSOS-201891

Title: Control theory helps to resolve the measles paradox

Response to Associate Editor Comments to Author (Dr Robert MacKay)

Reviewer's comment 1. *The paper looks good now. The only thing is that one reviewer questions how useful Box 1 will be. Perhaps it would work better to make two boxes, one for biologists to explain in simple language how you use control theory, one for control theorists to explain how you apply it to biology? The non-trivial issue for both seems to me that you are not using control theory in the sense of designing a feedback to achieve a purpose, rather you are interpreting nature's solution as implementing a control strategy.*

Authors' reply. *We thank the Associate Editor for their time and positive feedback. We confirm that the Associate Editor has understood our paper correctly. We are not using control theory to design or implement a feedback control to achieve any purpose. We are interpreting nature's solution as implementing a control strategy. We acknowledge the concerns regarding Box 1. We have decided to have two boxes, since we agree with the recommendations of reviewer 1 and the Associate Editor. We hope that this new version meets your standards for publication.*

Response to Reviewer 1

Reviewer's comment 2. *This version of the paper is much improved. The additions to the discussion on pages 6,7,8, 9 did a good job of putting the mathematical results in the context of the natural histories of the two diseases being considered, and clarified the points of agreement with existing experimental findings. The statements on page 9 "T-cell immunity enforces measles clearance during the typical drop in total lymphocyte count" and "this has not been found in previous studies" further emphasizes the new contribution of this paper. The statement on page 15 "As this reduction in the number of measles-specific T cells does not cause the reachability condition to fail..." emphasizes the interpretive value of the analysis done in this paper. I think the authors addressed my primary concern that this section was short-changed and inadequate before.*

I'm not sure what I think about Box 1. I thought that the paper needed to be expanded in its tutorial either into control systems for biologists or biology for control systems. I think the above mentioned changes made this a much more interesting paper for control systems people, as it explained the biology very well. Box 1 is a good tutorial on sliding mode theory for people already familiar with control systems, I worry it may not be general enough for a mixed audience. The concept of an invariant manifold within a state space, for instance, or of a distance function $s_0(t)$, are not going to be commonly understood topics outside of the people who already understand them. I would recommend Box 1 be re-written at a higher (less detailed) level with the mathematical biology community in mind. Focus on the implications, refer to other good survey articles for the mathematical details.

Authors' reply. *We are grateful to the reviewer for their time and positive feedback. We are happy that our second version has been informative for you. We acknowledge the concerns of reviewer 1 regarding Box 1. We used the feedback from the reviewer 1 and the Associate Editor to create two boxes. The new Box 1 is written in higher level language for biologists to explain in simple language how we use control theory. Box 2 is the former Box 1, since it is already a good tutorial on sliding mode theory for people already familiar with control systems.*

Response to Reviewer 2

Reviewer's comment 3. *I suggest to accept the paper.*

Authors' reply. *We are grateful to the reviewer for their time and positive feedback.*